# Nurses Response to the Physical and Psycho-Social Care Needs of Patients with COVID-19: A Mixed-Methods Study

**DOI:** 10.3390/healthcare12010114

**Published:** 2024-01-03

**Authors:** Angela Tolotti, Loris Bonetti, Corina Elena Luca, Michele Villa, Sarah Jayne Liptrott, Laura Maria Steiner, Colette Balice-Bourgois, Annette Biegger, Dario Valcarenghi

**Affiliations:** 1Nursing Development and Research Unit, Oncology Institute of Southern Switzerland, Ente Ospedaliero Cantonale (EOC), Via Gallino, 12, 6500 Bellinzona, Switzerland; angela.tolotti@eoc.ch (A.T.); corinaelena.luca@eoc.ch (C.E.L.); sarahjayne.liptrott@eoc.ch (S.J.L.); dario.valcarenghi@libero.it (D.V.); 2Nursing Research Competence Centre, Ente Ospedaliero Cantonale (EOC), Viale Officina, 3, 6500 Bellinzona, Switzerland; michele.villa@eoc.ch (M.V.); lauramaria.steiner@eoc.ch (L.M.S.); 3Nursing Department Direction, Ente Ospedaliero Cantonale (EOC), Viale Officina, 3, 6500 Bellinzona, Switzerland; annette.biegger@eoc.ch; 4Ospedale Regionale di Lugano, Ente Ospedaliero Cantonale (EOC), Via Tesserete, 46, 6903 Lugano, Switzerland; 5Cardiocentro Ticino Institute, Ente Ospedaliero Cantonale (EOC), Via Tesserete, 48, 6900 Lugano, Switzerland; 6Ospedale Regionale di Bellinzona e Valli, Ente Ospedaliero Cantonale (EOC), Via Gallino, 12, 6500 Bellinzona, Switzerland; 7Neurocenter of Southern Switzerland, Ente Ospedaliero Cantonale (EOC), Via Tesserete, 46, 6903 Lugano, Switzerland; colette.balice-bourgois@eoc.ch

**Keywords:** COVID-19, nursing care, qualitative research, surveys and questionnaires, nursing, intensive care units, pandemics

## Abstract

The COVID-19 pandemic heavily impacted nursing care. This study aimed to understand which nursing interventions were instrumental in responding to COVID-19 patients’ needs by exploring the experiences of patients and nurses. In this mixed-method study with an explanatory sequential design, we involved nurses caring for COVID-19 patients in intensive and sub-intensive care units and patients. In the first phase, we collected data through a survey that assessed patients’ needs from the perspective of nurses and patients, as well as patient satisfaction. In the second phase, qualitative data were collected through interviews with patients and nurses. In the third phase, we extracted quantitative data from patients’ records. Our sample included 100 nurses, 59 patients, 15 patient records, and 31 interviews (15 patients, 16 nurses). The results from the first phase showed patients and nurses agreed on the most important difficulties: “breathing”, “sleep/rest”, and “communication”. Nursing care was rated positively by 90% of the patients. In the second phase, four themes were identified through the patients’ interviews: “my problems”, “my emotions”, “helpful factors”, and “nursing care”. Five themes were identified through the nurses’ interviews: “the context”, “nurses’ experiences and emotions”, “facilitators and barriers to patient care”, “nursing care”, and “the professional role”. From the third phase, the analysis of the clinical documentation, it was not possible to understand the nursing care model used by the nurses. In conclusion, nurses adopted a reactive-adaptive approach, based on experience/knowledge, pursuing generalized objectives, and adapting their response to the clinical evolution. In difficult contexts, nursing care requires a constant competent technical-relational presence at the patient’s bedside.

## 1. Introduction 

In its last report, the World Health Organization (WHO) reported over 772 million confirmed cases and over 6.9 million deaths globally due to the SARS-CoV-2 pandemic [1]. Although the WHO has officially declared the end of the pandemic, it is important not to forget the lessons learned for similar future crises.

Front-line nurses played a critical role during the pandemic by caring for those affected by COVID-19 [2,3]. Nurses experienced significant demands for sacrifice, in terms of personal risk, increased workload, and difficulty in maintaining the usual healthcare standards [4,5,6]. Healthcare workers are at high risk of short- and long-term mental health problems, which may impact on the quality of patient care [4,5,7,8].

Nursing is always focused on providing personalized and holistic patient care [9] and on identifying innovative strategies to ensure the best possible quality, even in extremely adverse conditions. The impact of the COVID-19 pandemic was tremendous in terms of patient management, organization of care, and communication with patients and their families. This required considerable adaptation on the part of nurses and health professionals [4,5,8,10,11]. 

COVID-19 patients were exposed to devastating emotional distress from fear, uncertainty, and anxiety, and health care staff had to provide significant support to address and support patients’ feelings, needs, and experiences. Studies of other infectious disease outbreaks have shown that the risk of depression, insomnia, posttraumatic stress disorder (PTSD), and mental health problems for patients is substantial [8,12,13,14,15,16]. 

To ensure public safety, patients’ family members were not allowed to enter the hospital. Hospitals and health services had to rapidly adapt their policies and procedures to allow regular communication between patients and families despite the restrictions. Digital solutions were introduced, which were essential for family-centered care [17,18].

Nurses’ and physicians’ experiences during previous epidemics (severe acute respiratory syndrome (SARS), Middle East respiratory syndrome (MERS), haemagglutinin type 1 neuraminidase type 1 (H1N1), EBOLA virus disease) have been examined in several studies, which focused on the stressors that occurred both during and after the pandemic [4,19,20,21,22,23]. However, although the experiences of patients and their families are important to explore and understand, to address their needs [24,25,26], there is a paucity of literature on these aspects. Further findings may help to develop specific nursing interventions to better meet the needs of patients and their families [27].

In addition, little research has been conducted on what nursing interventions were implemented and to what extent they met the needs expressed by COVID-19 patients and their families.

Therefore, more in-depth knowledge about these variables would be useful in the event of new pandemics, to understand what type of nursing interventions might be implemented, what aspects are effective or ineffective, and why, in order to identify areas of best practice and enable appropriate improvement strategies [7].

Given the important role that nurses play in public health emergencies, it is essential to know which nursing interventions helped to ensure appropriate care and improved patient outcomes. Therefore, the aim of this study was twofold: to gain a better understanding of the nursing care provided in response COVID-19 patients’ care needs, and explore patients’ and nurses’ experiences during COVID-19 pandemic.

## 2. Materials and Methods

### 2.1. Study Design

This is a mixed-method study with an explanatory sequential design [28], consisting of a questionnaire to obtain quantitative data and semi-structured interviews to obtain qualitative data. We chose a mixed-methods design because it ensured a better understanding of the complexity of the healthcare environment during the COVID-19 pandemic [28]. 

This study was conducted from September 2020 to May 2021. Quantitative and qualitative data were analyzed separately and then triangulated to gain a more comprehensive understanding of the phenomenon, by means of joint display. The study protocol has been published elsewhere [29]. Unlike the original protocol, in this paper we report the results considering only patients’ and nurses’ data. Family members’ experiences will be reported in another paper, because they focused only on their own experience during the pandemic and not on nursing care. Therefore, their data are not consistent with the aims of this report. A description of the study phases is shown in Figure 1.

#### Context

The study was conducted at the Ente Ospedaliero Cantonale, the largest multi-site public hospital in the Canton Ticino region, serving a population of 350,000 people. The Ente Ospedaliero Cantonale includes eight hospitals, geographically distributed in such a way as to cover the entire Canton. Two hospitals out of eight were dedicated to the care of patients affected by COVID-19 during the first and second pandemic waves. Because of the rapid increase in numbers of COVID-19 patients, new beds in intensive care units (ICUs) were added and new ICUs were created.

### 2.2. The Quantitative Study Phase

#### 2.2.1. Participants 

The participants were nurses and COVID-19 patients, as described below.

##### Nurses

All Nurses (*n* = 560) working in COVID-19 sub-intensive medicine wards or intensive care units during the COVID-19 first and/or second waves were invited to participate in an online survey. We did not calculate the sample size because we invited the entire eligible population, based on the inclusion criteria (having worked in COVID-19 sub-intensive medicine wards or intensive care units during the COVID-19 first and/or second waves).

##### Patients

Eligible patients (*n* = 160) were identified through hospital databases of patients admitted to COVID-19 wards at the multi-site hospital during the pandemic. For patients, we also did not calculate the sample size because we involved the entire eligible population, based on inclusion and exclusion criteria.

To be included in the study, patients had to be 18 years or older, have contracted COVID-19, and have had a medium to high level of severity of the disease, defined as having been transferred to sub-intensive medicine wards or to intensive care units; patients also had to have been discharged from the hospital. Participants had to be able to answer a questionnaire and take part in an interview in Italian.

Patients with cognitive disorders or having an insufficient knowledge of Italian language were excluded from the study. In addition, individuals who were unable to provide written informed consent to participate in the study were excluded. 

#### 2.2.2. Data Collection

For data collection we used both self-made questionnaires and semi-structured interviews, as described below. 

##### The Nurse Questionnaire

To investigate nursing care for COVID-19 patients, a self-reported structured ad hoc questionnaire was developed and tested by involving nurses who had provided care to patients in COVID-19 units. The questionnaire items were developed considering the 11 needs of Cantarelli’s Nursing Performance Model (NPM) [30] (Table 1). 

The questionnaire also assessed nursing interventions during the pandemic and their perceived importance, using a ten-point Likert scale (where 0 = not important; 10 = very important). In addition, demographic and professional data were collected. A full description of the questionnaire is available in the original protocol [29].

The questionnaires were emailed to nurses eligible for the study. Through the questionnaire, respondents were also asked if they were willing to be interviewed. Data were collected at the end of the second wave of COVID-19, between September 2020 and May 2021. To avoid duplicate entries, respondents could submit their responses to the survey only once.

##### The Patient Questionnaire

To collect patient data, an adapted version of the nurses’ ad hoc questionnaire was used. Patients were asked to rate the level of importance of the listed needs, excluding diagnostic and therapeutic procedures (Table 1). The patient survey also assessed the nursing interventions provided during the pandemic and their perceived importance from the patients’ point of view. In addition, the Newcastle Satisfaction with Nursing Scale [31,32] was used to evaluate patients’ satisfaction with the care received. The Newcastle Satisfaction with Nursing Scale showed a good internal consistency reliability during cultural and linguistic adaptation in Italian (patients experiences of nursing care, α = 0.949; patients’ satisfaction with nursing care, α = 0.954) [31,32]. Before being administered, the questionnaire was checked for clarity and comprehension by five patients. 

A full description of the questionnaire items is reported in the original protocol of the study [29].

Patients invited to participate in the research were reached by sending them the questionnaires by post, along with the informed consent and a self-addressed stamped envelope. Patients were invited to return the completed questionnaire in the envelope provided and to indicate whether they were available to be interviewed.

#### 2.2.3. Data Analysis

The questionnaires were analyzed using descriptive statistics. Categorical variables were summarized using frequencies and percentage (%), with continuous variables using means and standard deviation (SD) or median (interquartile range (IQR)). The patients’ and nurses’ results were compared in a descriptive manner. IBM SPSS Statistics 26.0 software was used to conduct these analyses.

### 2.3. Qualitative Study Phase

#### 2.3.1. Qualitative Approach

For the qualitative phase, we used a qualitative descriptive approach. Semi-structured interviews were conducted with both nurses and patients, to gather in-depth insights about their experiences and about the phenomenon itself. The interviews followed a thematic structure based on the questionnaire and were conducted in a location agreed upon by the participants.

#### 2.3.2. Characteristics of the Investigators and Reflexivity

The researchers who conducted and analyzed the interviews were all experts in qualitative research. To guarantee reflexivity, researchers maintained a reflective diary. The interviews were independently coded by two researchers and, in case of disagreement, a third researcher was consulted to reach consensus. The researchers did not know and had not met the participants before the interviews. 

#### 2.3.3. Participant Recruitment and Sampling

For the interview, we used a purposeful sample. The number of interviews was determined using the principle of data saturation [33]. The guides with the questions for the patient and nurse interviews are reported in the Appendix A.

#### 2.3.4. Nurse Interview Recruitment

At the end of the online questionnaire, the respondent was asked if he/she was willing to participate in an interview. If the nurses agreed to participate, they were invited to contact the research team to schedule an interview, with the contact information reported at the end of the questionnaire. All the interviews were conducted in hospital or in participant’s homes, considering hygienic standards (hand disinfection, mask, and distance), in a quiet environment that guaranteed privacy and no distractions.

#### 2.3.5. Patient Interview Recruitment

The research team contacted the patients that were willing to be interviewed using the returned module sent by post. The interviews were then arranged accordingly. All interviews were conducted in the participants’ homes, considering hygienic standards (hand disinfection, mask, and distance). During the qualitative data collection, visiting restrictions were no longer active in our country.

#### 2.3.6. Data Analysis

The qualitative interview data with nurses and patients were analyzed using thematic analysis, as described by Braun and Clarke [34]. 

The material was transcribed verbatim, and NVivo 10 software was used to provide an overview and facilitate a systematic approach to analyzing the material. Three experienced qualitative researchers of the research team independently read the transcripts multiple times to become familiar with the data and gain a general overview of the text. Then the data were compared and reviewed for content. Any coding inconsistencies between the researchers were then discussed until consensus was reached. 

#### 2.3.7. Rigor Qualitative Phase

To ensure the rigor of the qualitative data analysis, the criteria of credibility, transferability, dependability, confirmability according to Guba and Lincoln [35], and fittingness according to Carnevale [36] were followed.

In the mixed analysis, we used joint displays as described in the literature [37,38,39] to bring the data together in a visual way and gain new insights beyond the information provided by the separate quantitative and qualitative results.

### 2.4. Analysis of the Nursing Documentation

The nursing documentation of patients who had been interviewed was also analyzed. It was considered an additional source of data, in addition to and compared with the results of the other sources (interviews and questionnaires). To improve the reliability of data extraction from patients’ records, two researchers extracted the data independently and then compared their results to check the consistency. Data extraction from the patient clinical records was performed according to the sections of the questionnaire, considering the needs of Cantarelli’s Nursing Performance Model [30]). 

### 2.5. Quantitative and Qualitative Data Integration and Rigor in Relational to Mixed Methods Research

To integrate the qualitative and quantitative data, we used joint displays, as described in the literature [37,38,39]. Joint displays are a way to “integrate the data by bringing the data together through a visual means to draw out new insights beyond the information gained from the separate quantitative and qualitative results” [37,39]. 

To guarantee rigor, based on mixed methods literature, we used the legitimation criteria described by Younas et al. [38,40] (Appendix A).

## 3. Results

### 3.1. Quantitative Phase—Nurses and Patients Surveys

The surveys with patients and nurses were conducted in July 2020. Of the 560 nurses invited, 100 (18%) responded to the survey, whereas of the 160 invited patients, 57 (35.6%) responded. Of the nurses, 47.9% were aged between 31 and 50 years, 49% had between 6 and 20 years of work experience, and 34.7% had worked in an ICU during the pandemic.

The patients had a mean age of 69 ± 11 years, were mostly males (69.5%), and generally with a good level of autonomy with daily life activities (93.2%); 54.2% were affected by at least one disease, and the most frequent comorbidity was arterial hypertension (28.8%). The median length of stay was 17 days (13–37) (Table 2).

The nurses indicated, always in order of importance, the need for breathing (Median = 10 pt. (25th percentile–75th percentile) = (9–10)), the need for communication (10 pt. (8–10)), and the need to maintain cardiovascular function (10 pt. (8–10)). With regard to their care needs, patients reported, in order of importance, the need for breathing (Median = 6 pt. (25th percentile–75th percentile) = (3–8)), the need for rest and sleep (4 pt. (2–8)), and the need for communication (4 pt. (1–8)). On the other hand, regarding the assessment of the needs for treatment and diagnostic investigations, for which only nurses were surveyed, 9 pt. (8–10) were obtained for the first and 8 pt. (7–10) for the latter. Although nurses and patients provided different assessments with reference to the nine needs explored, two of the first three needs were in the same order of importance (Figure 2).

Table 3 shows the nursing interventions that, according to the perceptions of nurses and patients, had been implemented most and the relative degree of utility attributed to them, with a score ranging from 0 to 10 points. Even in this respect, there were some differences between the evaluations made by the two groups, with regard to the need to breathe, for example, patients reported that the “administering oxygen” was the most frequent and important intervention, reported in 85.9% of cases (49/57), with a degree of importance of 8 points out of 10 (5–10). Instead, for the nurses, the most frequent intervention in the context of this need was “assessment and surveillance of respiratory parameters”, reported in 99% of the cases (99/100) with a degree of importance of 9 points (9–10). For the other needs, such as communication or nutrition and hydration, the interventions considered priorities by both groups were largely the same. 

Nurses were also asked which new interventions they had to implement to address patient needs during the COVID-19 pandemic. 

Table 4 shows, for each need, the most frequently indicated new intervention. Among these, the most reported were pronation in 13% of cases (13/100), the adoption of innovative communication strategies such as video calls in 10% (10/100), and the use of devices for oxygen (Venturi system, CPAP) in 7% of cases (7/100).

From the responses given by the patients to the questionnaires, the overall evaluation of nursing care received was in 91.5% of cases (54/57) considered as “good/excellent” with a median value of 6 (5–7) on a scale from 1 to 7 points. The overall assessment of their hospitalization in the COVID-19 ward was considered good/excellent in 89.8% (53/57) of cases, with a median value of 6 (5–7).

It is worth highlighting that the aspect patients appreciated most with respect to the nursing care received was having been considered as persons and having been treated as such, ensuring them the due confidentiality and decency. Some criticisms emerged instead regarding the limited amount of freedom granted in the ward, the amount of time that passed between a patient’s call and the arrival of the nurse at the bed, and the way in which the nurses reassured relatives and friends. 

Overall, most of the items that we investigated were rated very positive (Figure 3).

### 3.2. Qualitative Phase

#### 3.2.1. Nurse Interviews

A total of 16 interviews were conducted between October 2020 and March 2021, lasting between a minimum of 36 min and a maximum of 67 min. The sample included nurses who worked in COVID-19 intensive and sub-intensive medicine wards at the hospital where the study was conducted. The participants were 7 male nurses and 9 female nurses, aged between 25 and 56 years (average age: 38.6 years).

Following the thematic analysis of the nurse interviews, 5 macro-themes were identified: “the context”, “nurses experiences and emotions”, “facilitators and barriers to patient care”, “nursing care”, and “the professional role” (Table 5).

#### 3.2.2. Macro-Theme: “The Context”

The context in which nursing care took place was difficult from several points of view. This macro-theme groups together two themes: “the clinical/professional context” and “the organizational context”.

#### 3.2.3. The Clinical/Professional Context

The difficulties regarding the context from a clinical point of view included the initial lack of knowledge about the disease and its evolution, the sudden changes in the clinical situation, and the frailty of the patients.


*“…they were fine until that moment, so they even talked to you, but maybe after only half an hour they were dead. So, also this unpredictability, this way of dying was so rapid, that you didn’t have time to realize it…”*
(Nurse 6)

From a professional point of view, the context was difficult due to the heavy workloads, emotional involvement, difficulties communicating with patients, lack of privacy, situations of ethical conflict, high mortality rates, and facing new and potentially dangerous situations.


*“…giving each other a hug, which in theory okay you couldn’t, but there many times someone did it anyway and I was never able to say no, because in any case you see a mother, 90 or 80 years old, with her daughter, they haven’t seen each other, she’s sick, a hug, they’re wearing a mask, so hug each other, her overcoat is lifted up, it’s okay, hug her, if she’s already in there, for me there’s no problem.”*
(Nurse 1)

#### 3.2.4. The Organizational Context

From an organizational point of view, the increased number of beds in the intensive care unit, the setting up of multi-professional teams, the lack of some materials (in the initial phase) were identified. In particular, there was a transmission of intra-professional expertise between nurses with different clinical experiences, depending on which patient problems had to be managed. Another collaboration that was perceived to be important was that with physicians, as even if there was not much information about the disease, there was involvement in discussions and mutual professional appreciation.


*“…the collaboration between the various services […] with palliative care, even the relationships with the doctors were different, the doctors were from outside the hospital, so they had other ideas, other ways of helping each other, and then the support between us was fundamental.”*
(Nurse 5)

### 3.3. Macro-Theme: “Nurses’ Experiences and Emotions”

This macro-theme groups together six themes: “you won’t believe it until you see it”, “it was a nice experience”, “your own emotions”, “confrontation with death”, “your own resources”, and “I no longer want to be a nurse”.

#### 3.3.1. You Won’t Believe It until You See It

Confrontation with reality was psychologically traumatic, nurses were not initially prepared to manage the situation. Even just seeing so many patients intubated and sedated in large rooms was already impressive on its own, as were the faces deformed by pronation. The following emotions emerged: (a) perceiving fear in the patients’ eyes and their requests to be saved; (b) observing the rapid evolution of the disease and the many deaths in solitude due to the absence of their loved ones; (c) experiencing a profound sense of helplessness in the face of all this, and with the fear of not knowing how to do the right thing.


*“What I remember most, it’s bad to say but those faces deformed by pronation, prolonged pronation and then damaged faces, swollen faces, that left a bit of an impression on me.”*
(Nurse 15)

#### 3.3.2. It Was a Nice Experience

The experience was seen positively and as enriching from a professional and personal point of view. It helped in becoming mature and changed perceptions of what was truly important. The first phases of the experience were considered interesting and nice, despite the dramatic situation.


*“I think I’ve grown up a bit, I can talk for forty minutes without crying; I’ve grown a little, maybe I’ve learned how to better process the most critical situations and challenge myself more, and to start again as if I didn’t know anything.”*
(Nurse 9)

New colleagues were warmly welcomed, and it was a positive experience to work together with a common goal, however over time, enthusiasm, professional spirit, and availability waned, and everyone became tired. After the first wave, the support of the population also diminished.


*“if we managed to do what we did during the first wave, it was also because we felt the support of the people who were outside, and I believe that this tiredness we have now is also because you feel that there isn’t any more support.”*
(Nurse 7)

#### 3.3.3. Your Own Emotions

The fear of this almost unknown new virus was great, as was the fear of infecting one’s family, marking the nurses’ daily lives in those days.


*“(…) I was afraid of infecting my family; in fact, I didn’t see my parents and my in-laws, who are old, in their eighties, for four months.”*
(Nurse 11)

The solitude of patients both when alive and dying was experienced as emotionally touching and it was hard to sustain some interpersonal relationships, especially if they were prolonged. In the ward, very strong and different emotions were experienced based on the outcome of the disease for the patients, with nurses reporting how much they suffered when they saw patients dying. Nurses clearly remembered those stories that had a profound impact and struggled to talk about them. They said that it was like taking something out of a drawer that they wanted to keep closed to protect themselves.


*“Not thinking what’s outside. Once the doors are closed, I completely disconnect, in fact I still can’t talk about it very easily today, a bit of that experience remains; I try to keep it in a corner.”*
(Nurse 10)

#### 3.3.4. Confrontation with Death

The idea of death hovered around the caregivers, to which they reacted with contrasting behaviors. On the one hand, to protect themselves, they became insensitive to the many patient deaths and focused on technical actions. Healthcare activities in that case became a set of procedures that needed to be carried out, as if it was an assembly line. Being close to a dying patient could also be a profound need of the nurse.


*“… She died in the afternoon, I was looking after her that morning; it was nice anyway, she didn’t suffer and so this is what counts most for me…”*
(Nurse 1)

For some it was also difficult to accept the order in the department that no critically ill patient should be resuscitated. A disposition against which one nurses rebelled, even if they knew that there was no chance of obtaining a result.


*“Instead of crying, this made me feel angry, and with some feelings of rebellion, I said: “I’ll resuscitate her anyway, if it’s a colleague who has a heart failure, if it’s you I’ll resuscitate her anyway.”*
(Nurse 11)

#### 3.3.5. You Own Resources

A strong motivation, even stronger than fear, was the desire to get involved and help others. Nurses were therefore immediately available, driven by a mix of ethical-social and personal-professional motivations. What emerged was the awareness of the nurses’ social role and the desire to contribute to the situation of emergency through their professionalism in the front line.


*“the fear was there, but the urge and the willingness to go and take part in this experience, and lend a hand was even stronger (…)”*
(Nurse 1)

Being expert nurses was also useful to improve yourself in this situation and identify strategies to manage one’s emotions on the field.


*“Experience certainly helps, it has helped and continues to help a lot.”*
(Nurse 8)

Having constant communication and support from colleagues contributed to overcoming difficulties.


*“….super supportive colleagues, during the night, when one is having a moment of discouragement, there was immediately another one who was ready to listen to and understand the other person’s experiences and feelings.”*
(Nurse 10)

#### 3.3.6. I No Longer Want to Be a Nurse

The emotional stress nurses felt was considerable, to the point of thinking of abandoning the profession.


*“(…) for two weeks I was already on the verge of saying “no, I’ve had enough, I give up, I’m stopping, I don’t want to be a nurse anymore”, but what’s the point of being a nurse in this way?”*
(Nurse 10)

### 3.4. Macro Theme: “Facilitators and Barriers to Patient Care

This macro-theme includes five themes: “interprofessional collaboration”, “support from superiors”, “being experts”, “lack of specific competencies”, and “standardization of care”.

#### 3.4.1. Interprofessional Collaboration

The COVID-19 emergency led to the creation of extemporary teams of professionals with different levels of competence and experience and who in part had never worked together before. The great and inevitable initial difficulties were overcome by the climate of discussion, collaboration, and reciprocal support that was immediately established in most of the teams. There were lots of discussion with physicians about both the situation and the measures in force.


*“I found an exceptional team and colleagues, there really was incredible solidarity, a spirit of collaboration, everyone really united for one purpose, which was to face this pandemic.”*
(Nurse 11)

#### 3.4.2. Support from Superiors

The support from local management (department heads) helped to manage the situation in a much better way.


*“He was just asking “how are you?” or the department heads, who were also there, three department heads who managed to collaborate together, would arrive and ask “how are you?”*
(Nurse 7)

#### 3.4.3. Being an Expert

Having previous experience in intensive care units and knowing how to make competent and informed decisions was of great help. The different disciplinary background of the nurses was experienced as both a professional and human resource.


*“I believe I have given my small contribution as a person who has worked with critically ill patients for more than twenty years, who however knows how to recognize certain changes, knows how to anticipate, perhaps even situations that may evolve in a negative way.”*
(Nurse 7)

#### 3.4.4. Lack of Specific Competencies

Nurses were faced with the need to support specialist areas, albeit without having specific competencies and without an adequate period of coaching/support. In that situation, nurses tried to learn quickly by observing and asking for continuous exchange of opinions with colleagues who were experts in critical care.


*“people who did not have training supported those who worked directly on the patient or in any case if they did something on the patient they were directly supervised, controlled and evaluated by those who were responsible for the patient.”*
(Nurse 8)

#### 3.4.5. Standardization of Care

The main difficulties nurses mentioned were having to integrate their own working methods with others, the need to reduce the quality standards of care due to the heavy workloads, and the standardization of the activities to be carried out.


*“During the technical phase we are all a bit like robots, so we just go in with it, we know what we have to do and we do it, as I was saying, in a standardized way.”*
(Nurse 15)

### 3.5. Macro-Theme: “Nursing Care”

Four themes were grouped into this macro-theme: “the nursing care approach to COVID-19 patients”, “assessment of patients and their needs”, “professional objectives and priorities”, and “professional performance”.

#### 3.5.1. The Nursing Care Approach to COVID-19 Patients 

In this heterogeneous and unstable situation, the nursing care approach identified through the interviews appeared very varied; being more responsive in some cases, and mostly planned in other cases.


*“…I set myself smart goals, I have to achieve them in a certain amount of time in order to have a tangible result.”*
(Nurse 11)

Nurses tried to have general overview of the patients in the ward and each time decided the intervention priorities. The aim of this approach was to keep various needs/problems under control, considering the complexity and frailty of their patients.


*“In reality our approach is a global approach, in the sense that we were not just focused on one thing, the real problem was breathing, but in reality, there are all the other things that are part of our care.”*
(Nurse 13)

#### 3.5.2. Assessment of Patients and Their Needs 

In the acute phase of the disease, the clinical condition could change dramatically in a very short time, and the nurses used a reactive-adaptive approach, based on their experience and knowledge.


*“a precarious balance … dictated by the fact that this virus was unknown, there were no data, it was not well understood how it behaved and consequently how we should behave.”*
(Nurse 16)

Nurses reported that their difficulty in understanding the sudden change in patients’ clinical condition was also due to the patients’ lack of perception that their respiratory condition was worsening.


*“…they didn’t always realize that they were struggling, so it wasn’t always a need that they expressed, or they started to express it when it was already a bit late.”*
(Nurse 9)

Generalized objectives were pursued (i.e., keeping patients alive, avoiding suffering), adapting the response based on clinical evolution and new knowledge.


*“For us it was really important to understand what our patients needed, because it was very important to help them at that moment, in total solitude, in total discomfort, it was a priority to understand what they needed… we just couldn’t understand what the need was; this caused enormous distress for patients and enormous distress for staff.”*
(Nurse 10)

The priority assessment activities were aimed at controlling and supporting breathing and vital functions. Respiratory weaning was one of the most relevant care problems and required great efforts and solid competencies, typical of an intensive care nurse.


*“It’s not that I just turn the knobs of the ventilator, I turn patients on their hips, I auscultate them to understand if there are any secretions, I understand if inhalations are needed, I understand if some positions help them.”*
(Nurse 11)

In the clinical nursing assessment of the patients, respiratory rate became the main and most important parameter, defined by the nurses themselves as a paradigm shift.


*“Generally, we almost never assess respiratory rates, but more or less now this happens in all hospitals, here this was the first sign, the first of all it was the respiratory rate, together the temperature. So, also your way of viewing nursing changes.”*
(Nurse 3)

#### 3.5.3. Professional Objectives and Priorities

Professional objectives were often present and stated in very general terms. In some cases, they were described in more detail and related to specific professional interventions. Some of the most common objectives included dealing with patients’ problems, supporting vital functions, addressing their needs, promoting functional recovery as much as possible, making them feel well, promoting contact with their families, supporting them with their relations, not letting them die, or looking after them in the final stage of their life.


*“let’s set ourselves a goal, we have to extubate him within four days, it’s a goal we set ourselves; we then realize that we can’t? Well, okay, let’s talk about it again, but let’s set real, achievable objectives.”*
(Nurse 10)

On other occasions, the objectives, although still generic, had very limited timeframes, such as keeping patients alive during their work shift or helping them to rest during the day. The general objectives, often not explicit or written, offered a general guidance to professional actions. Once the acute phase had been overcome and the vital signs were stable, treatment objectives and problems evolved in favor of more personalized care.


*“it was a bit as if we were at war, so the important thing was to go straight to the point and making sure that patients got well.”*
(Nurse 11)

#### 3.5.4. Professional Performance

Relationships and communication were carefully addressed by the nurses at various times; in the acute phases, to make patients feel comfortable and reassure them, to support them during the phase of respiratory weaning, to contain them during acute states of confusion after awakening, and to reassure them during their functional recovery process.


*“…there was someone who was also looking for a presence, simply having someone close by, wanting to talk, keeping you there to talk, or trying to talk, to let you know something, to know something about their family, in my opinion this was what they were looking for mostly.”*
(Nurse 8)

Having more time to spend with patients was welcomed by nurses when it was possible to do so. With conscious and intubated patients, effective communication was complex, and with the use of different means (e.g., lip reading, use of cards, writing, telling about oneself or about one’s journey during hospitalization).


*“Speaking, for example, with a tracheotomized, ventilated patient is, in my opinion, one of the most difficult things, in the sense that the movement of the lips does not always correspond to what someone actually wants to say to you, so […] sometimes it’s frustrating because maybe the patient is telling you something and you make various attempts trying to understand what he’s telling you…”*
(Nurse 13)

The relationship with patients was important. Nurses tried in different ways to encourage contact and communication between patients and their family members, and communicating via video calls was appreciated when it was possible. The nurses’ relationships with family members were generally occasional and occurred in particular situations because, due to the sensitiveness of the situation, communication with the patients’ relatives was entrusted to physicians, a choice that nurses experienced in an ambivalent way.


*“…this lady in the end had been treated for more than a month and then seeing her again after she recovered, the first contacts with family members, and the video-calls with family members was something that left you speechless.”*
(Nurse 8)

The theme of the humanization of care and respect for dignity emerged as a priority experience for many colleagues. The desire to bring daily practice back to the original values of caring was perceived. Experiences of ethical conflict were expressed in various situations: when one wonders what is right to communicate to patients and/or their families, when one decides to interrupt treatment, when you are unable to dedicate to a patient all the time they needed, and when choices made by physicians were not shared.


*“maybe they die like this because I’m not capable of… and instead they say to you “look, it’s not your fault, but it’s just the situation that…” so if not in a normal case of accompaniment you maybe have time, maybe you have a way to… you didn’t have it there and you had to quickly understand what to do to not make them suffer.”*
(Nurse 5)

The new disease and the work context required the acquisition of new knowledge and skills. For intensive care nurses, no entirely new nursing techniques were performed, but many were performed under difficult conditions and very frequently. 

The new activities involved the management of certain devices (such as the Venturi system) and the use of new tools for the evaluation of the clinical evolution of patients, such as the early warning score and the Ti-Cos score for the clinical evaluation of patients, and to evaluate the evolution of symptoms and possible disease trajectories (from conservative to maximal treatments). Other situations experienced as new included having to supervise the activities of less experienced colleagues, despite having little experience in the assigned ward.


*“Even for the vital signs, taking the respiratory rate is not something we all did every day, so I also had to get back into it again. Okay, I really have to count the breaths, understand… that is, they are also new things that we didn’t do before; the use of the Venturi, that was also new, so even the healthcare assistants didn’t know it, so they turned up the oxygen, but they didn’t know what they were doing, so we also had to supervise a bit.”*
(Nurse 6)

#### 3.5.5. The Outcomes of Nursing Care

The outcomes of nursing care were not described by nurses in a specific and timely manner. They described them in general terms, such as absence of complications, maintaining alive, regaining functional autonomy, and being thanked by patients and family members.


*“…a lot of patients thanked you, so that was nice, because they had also understood our level of stress.”*
(Nurse 1)

### 3.6. Macro-Theme: “The Professional Role”

This macro-theme groups together two themes: “a multifaceted role” and “fundamental care”.

#### 3.6.1. A Multifaceted Role

The professional role of nurses is multifaceted: they must be able to anticipate and/or capture patients’ potential problems early, help them re-orient themselves in the awakening phase, support them, and make them aware of the small steps of their progress. Nurses must have interpersonal knowledge and skills.


*“among our skills which, in my opinion, which is perhaps one of the most important parts, is that of communication, of the relationships… because when a patient is going through weaning and is making it, and therefore the supports offered by machines begin to be a little less important, the difference that emerges a little is the communicational-relational part.”*
(Nurse 13)

#### 3.6.2. Fundamental Care

The patient’s autonomy in self-care is considered important, as well as certain actions or gestures of care (fundamental care). The nurse brings added value by being competently close to patients for many hours every day, making them feel their closeness and respect.


*“in my opinion we, needed to go back to the origins a bit, in terms of treatment […] it’s not that I’m not for protocols, protocols are fine, I’m for rules, but every now and then we need to go back to the basics a little more.”*
(Nurse 3)

### 3.7. Patient Interviews

Fifteen interviews were performed between October 2020 and March 2021, and 15 patients were interviewed, inlcuding 4 women and 11 men aged between 53 and 79 years (average 70.8), with an average length of stay of 40 days in intensive and sub-intensive care units. 

The interviews took place at the patients’ homes and lasted between 24 min and 1 h. From the analysis of the patient interviews, four macro-themes were identified: “my needs/problems”, “my emotions”, “what helped me”, and “nursing care”. Specific themes were identified for each macro-theme (Table 6).

#### 3.7.1. Macro-Theme: My Needs/Problems

This macro-theme groups together five themes: “I had no information or it was incomplete”, “putting all the pieces of the jigsaw puzzle together”, “physical problems”, “psychological problems”, and “communication/relational problems”.

#### 3.7.2. I Had No Information or It Was Incomplete

This theme is mainly linked to the information received from patients regarding the need to be intubated. This information, due to the sudden worsening of clinical conditions, was sometimes given in a very short time and this did not allow the person to process the information, often generating confusion and distancing them. When patients were asked what they intended to do if intubation became necessary, they experienced a feeling of helplessness, which often occurred when they perceived the seriousness of their condition for the first time.


*“even later they told me “he must be intubated”, but I didn’t understand anything, I understood that they had to intubate me, but I… and then I understood that they said “no, but before doing so we have to inform the relatives at home and everything”, I couldn’t see anything, I was practically alive but always with an empty brain, not thinking about anything, and then they intubated me.”*
(Patient 12)

Patients reported that they did not always have the necessary information regarding the time of awakening. Not knowing why they wake up intubated or unable to walk has led them in some situations to experience strong feelings, such as fear, anguish, or experiencing unreal situations.


*“that’s what they didn’t explain to me; if they had explained to me as soon as they woke me up and told me “you can’t talk because we intubated you, you can’t walk because you’ve been sedated, tomorrow you will try, you will start walking”, then maybe I would have felt less anxious.”*
(Patient 15)

#### 3.7.3. Putting Together All the Pieces of the Jigsaw Puzzle

When they woke up, patients often struggled to understand where they were and told of their need to reconstruct the missing information, to understand what happened and what was happening. The explanations provided by the nurses with respect to the nursing actions they were going to perform allowed them to cope better with some uncomfortable situations, such as the presence of tubes and probes. When faced with a lack of information, patients often did not attribute this responsibility to those who cared for them, but to themselves, believing that they should have asked more questions.


*“…because then even when you are back in the ward you start putting together all the pieces of the jigsaw puzzle, you receive all the various news. And again, long sleepless nights, I think that probably started sleeping in the last few days… and then automatically also…, and there all the pieces of the puzzle of everything that happened come together, it was an incredible thing, that is, it seemed as if you were in a film, a very unusual thing.”*
(Patient 2)

#### 3.7.4. Physical Problems

Patients talked about pain that sometimes involved the whole body, sometimes it was localized, of the tiredness that led them to only want to sleep and made them feel strange and that fever worsened this feeling. Difficulty breathing, sometimes also due to the presence of secretions, made people experience the fear of dying. Tracheostomy was narrated as the worst thing that can happen. Another problem that appeared in various narratives was the difficulty in falling asleep. In some cases, this led them to experience negative feelings, such as terror. The presence of a nurse helped them to manage their negative feelings.


*“The fever. Breathing, I breathed quite well. Afterwards, it was the fever that made you feel tired, the fever.”*
(Patient 11)

#### 3.7.5. Psychological Problems

Patients described their body image. Seeing themselves shabby caused discomfort and suffering. Another element of suffering was that time passed slowly and the presence of other people inhibited them from watching the television to distract themselves.


*“They were long, I didn’t shave for a fortnight, I had little hair, very little, I shaved my hair every week, I look old, aged… I showed them, I showed them the photo because I had myself photographed in the hospital…, and afterwards I said to myself “I’m ugly and old.”*
(Patient 7)

#### 3.7.6. Communication/Relational Problems

The need to wear masks made communication between patients and nurses difficult. It was difficult to recognize faces and sometimes seeing nurses covered with gowns, masks, and other PPE was intimidating, especially at night. Another element that patients described was the fear of disturbing the nurses with their requests. People talked about waking up and discovering that they could not talk because of intubation. This even caused feelings of despair.


*“I was down in intensive care and there was a young lady who said to me “try to say a word”, I couldn’t do it, well, this is what I remember, that I couldn’t speak. And after that she gave me a pen and paper and said “try it”, I couldn’t hold it, I couldn’t hold either the pen or the paper.”*
(Patient 1)

### 3.8. Macro-Theme: My Emotions

This macro-theme groups together three themes: “worry and fear”, “frustration and helplessness”, and “priorities that change”. 

#### 3.8.1. Worry and Fear

Patients expressed concern for their family members and a desire to tell them where they were and inform them about their conditions. Patients also talked about other people who died from the same disease. Sometimes they were other patients in the same room, and at other times their friends. Seeing and hearing what happened to others was frightening, even for their own fate, and the images broadcast on television increased this fear. Patients talked about the experience of seeing and hearing everything that happened to other patients in the same room. Patients also said that other patients in their rooms died after being intubated. Some also witnessed the deaths of other patients. This experience was described as shocking.


*“We could hear everything, and the doctor came many times a day with this gentleman “look, we have to intubate him”, “no no no…”. Then they told him “look, now let’s call his wife, the daughter of this gentleman” and they called them, but before calling them, the doctor came, told him “the last thing I can tell him is if he doesn’t want to be intubated he has three days left to live” and then his wife and daughter arrived and they talked, I don’t know, and they took him away, I don’t know if they intubated him, and after a couple of days like this, I wasn’t aware of anything anymore, I didn’t understand anything anymore, I didn’t react, I didn’t think about anything anymore, my brain was empty..”*
(Patient 12)

Even the external context was a source of fear: the presence of military ambulances, soldiers, and civil protection tents contributed to creating surreal, war-like scenarios. These scenarios made patients feel that something frightening, unimaginable was happening.


*“I didn’t know anymore… and then I thought that with all the movement that was down there, because it seemed as if we were at war, because I saw it, I saw these ambulances, the tents, curtains… it seemed to be a war, because the soldiers took me here in their ambulance, it was the whole environment, I’m saying something really horrible happened here…”*
(Patient 7)

Some people talked about feeling imprisoned, not only in relation to the war-like context but also due to the limitations imposed by the situation, such as only being allowed to move within restricted spaces and difficulties in communicating with the healthcare staff.


*“No, I don’t want to blame everyone. I was there in a room, I don’t remember if the room number ended with six or two, I don’t know that, there was the possibility to move along the corridor from the door of my room to the window; there was no balcony, there was a balcony window there, which you could open, but it had bars, it looked like a prison.”*
(Patient 8)

#### 3.8.2. Frustration and Helplessness

Patients also talked about their weakness and dependence on nurses, about no longer being able to do anything, experiencing a feeling of hopelessness and even thinking about death. Patients also talked about their feelings of embarrassment and the shame they felt in having to depend on the nurse even for personal hygiene but also how the nurses’ professionalism helped them not to feel embarrassed.


*“…when you are there in a room, also embarrassed by certain things you have to do, because all of a sudden you are naked in front of a woman, a nurse ok, but you are there naked, in short, and for me it is extremely important to have never felt humiliated…”*
(Patient 3)

#### 3.8.3. Priorities That Changed

This theme is related to when patients were told that they would be transferred to the intensive care. They felt the need to communicate with their family members, to say hello to them and give something of themselves to them. At that moment they talked about how their way of thinking changed, about the things that were important.


*“and then all of a sudden you’re in a situation where everything you thought about in the ambulance no longer has any importance. You are there and all of a sudden… you slip into a condition where all the material things, but unfortunately also affections, are no longer a priority… in that moment you slip, and you no longer think about anything or anyone… and all of a sudden you’re just focused on yourself, but not in a selfish way, it’s just that you can’t do anything about it…”*
(Patient 3)

### 3.9. Macro-Theme: “What Helped Me”

Regarding the macro-theme “what helped me”, internal and external factors were identified.

#### 3.9.1. Internal Factors

Noticing improvements in the activities that patients could perform helped and supported them, as did returning back home, seeing their family, and thinking about all the things they could still do, such as their hobbies.


*“The first thing is to go back home and see my family. Then I still have several things to do. So, I don’t want to die now, I have various works to put together, in my atelier, there in the garden; I still have various things to do, and I can’t die now…”*
(Patient 5)

#### 3.9.2. External Factors

Patients underlined the importance of maintaining contact with their family members by telephone. The nurses encouraged this communication by also offering video calls, which were greatly appreciated by patients and families. The possibility of having a roommate with whom you can spend time together and talk was considered very important. In a situation of pain and fear, hearing the care healthcare staff chat and sing gave patients hope, and made them think that life would continue normally.


*“They removed the tube, they put a microphone here, something to be able to make a video call, so that I could talk to my family at home. That was very good for me, I was happy, and that’s what I think helped me react most, the video call I made;”*
(Patient 5)

### 3.10. Macro-Theme: “Nursing Care”

This macro-theme groups together three themes: “caring activities”, “professionalism”, and “how nurses are perceived”.

#### 3.10.1. Caring Activities

Patients described how they appreciated some of the nurses’ acts of particular attention. Patients appreciated the explanations that accompanied the nursing actions. In particular, patients appreciated that nurses knew how to approach the body of another person. Patients talked about how they felt supported by the nurses through their delicate and attentive gestures.


*“Always, everything, even when they washed me, when they came to wash me, they explained to me: “here we’ll put this cream because we have another cream for your feet”, this nurse did everything… you can’t be any better than that.”*
(Patient 12)

#### 3.10.2. Professionalism

Participants described the competence of nurses through some elements, such as answering their questions, and the efficiency and safety of the activities they carried out. Another element they underlined was the humanity of the nurses. Patients felt cared for and felt that their clinical condition was looked after. Seeing their own improvement made patients feel well taken care of. The quick and timely responses to their needs were appreciated.


*“…yes, various times, this humanity was expressed in various way, it was expressed by just being there, by leaning over you, and also perhaps even extremely, on the one hand the virtual caress, but perhaps also just feeling a caress. People, especially older people, I think… have an incredible need to be touched, to have a caress, so if a nurse does that to you at a certain moment it’s like a positive electric shock, it’s joy; …”*
(Patient 3)

#### 3.10.3. How Nurses Are Perceived

The image that patients had of the nurses was very positive. Some patients underlined how they felt cared for and protected, and in this regard they described the nurses as their “guardian angels”. Regarding any suggestions for the nurses, patients highlighted their satisfaction with the care they had received and urged nurses to keep it up. Someone underlined the importance of continuity in the relationship, of being listened to, especially when patients were unable to sleep at night, as well as dialogue because they experienced situations of isolation.


*“Well, then after a while I didn’t see her anymore because she was moved to another ward… This happened all the time: you establish a certain relationship with this nurse, but then I didn’t see her anymore, and that… that was a bit of a shame.”*
(Patient 2)

### 3.11. Results from the Clinical Patient Records

The nursing process reported in the clinical records of the interviewed patients (*n* = 15) was analyzed, starting from the needs of the individuals, according to Cantarelli’s model [30], as well as through integrating the phases of the nursing care process (detecting the need/problem, identifying the objectives, identifying treatment interventions, and evaluation). The analysis of the documentation highlighted that the care process was fragmented. Compared to the needs identified, in most of the records, the treatment objectives were implicit or absent. However, we observed that, even if the objectives were implicit, the planned care interventions were related to the described need for care.

Regarding the identification of the patients’ needs, some were reported in all cases (e.g., breathing, elimination, and cardiovascular), other needs were reported more frequently (e.g., nutrition, hygiene, movement, safe environment, and communication). The need least represented was the need for sleep and rest (*n* = 3 cases). With regard to the procedures, the most frequently reported were about treatment, whereas diagnostic procedures were absent.

The explicit objectives were significant in terms of prevention and recovery of autonomy in the short-medium term, such as weaning from the ventilator and modulation of nocturnal ventilator support to promote daytime respiratory autonomy, as well as regaining autonomy and strength for daily life activities. The objectives expressed in these terms should guide the completion of nursing care actions.

Treatment interventions were divided into complex interventions, which required more in-depth clinical evaluation/judgement, and into simple, more routine, and standard interventions, which did not require complex clinical judgment. Among the interventions requiring this expertise, those most frequently reported were about the need for breathing and the cardiovascular system, considering the novelty of the pathology and incomplete knowledge of its clinical evolution. The evolution of the disease in some cases highlighted a sudden deterioration in the clinical situation, which probably required greater clinical judgment skills for early identification of clinical situations that could rapidly evolve towards instability. Among these actions, the following were frequently reported: management of oxygen flows and respiratory devices, prono-supination for respiratory management (an activity not frequently applied before the COVID-19 outbreak in intensive medicine, but which became a frequent activity with a high consumption of resources), and continuous monitoring of vital signs, particularly respiratory rate.

Another intervention, which however required professional communication skills, was the management of crisis conversations with family members and patients, as well as managing communication problems with conscious patients with cannula and speaking valves. No specific skills were required, but previously acquired skills needed to be stressed.

Evaluations of care activities were mainly periodic and related to checking the actions undertaken. It was observed that the final evaluations were directly related to the actions implemented. The final evaluations related to the explicit and defined treatment objectives were poorly described, mainly because these objectives had not been explicitly defined.

Among the specific actions initiated for patients with COVID-19 was the use and management of the early warning score scale (which is very sensitive in defining patients at risk of clinical instability), and in only one case was there a consultation with a spiritual assistant, to manage a crisis of existential/spiritual values. A further action, which became a standard treatment for these patients, was the activation of respiratory physiotherapy.

Consistency of the implemented care process appeared to be lacking, especially the part regarding the definition of the objectives in an explicit way and, consequently, during the evaluation of the outcomes and the achievement of the defined treatment objectives. From the analysis of the clinical documentation, it was not possible to understand the nursing care model used by the nurses. It did not appear that the nursing care process was guided by a precise theoretic reference model but tended to respond to the needs identified day by day.

A full description of the data extraction from patients’ records is reported in Appendix A.

### 3.12. Integration of the Qualitative and Quantitative Findings

The joint display that integrates the qualitative and quantitative findings is reported in Table 7.

Consistency emerged between person-centered care being referred to by nurses and the level of patient satisfaction. Indeed, 80% of the patients were fully satisfied with the care they received and 78% of the nurses were aware of their needs. The nurses put great effort into ensuring the highest quality of care, and the patients realized this.

During the clinical nursing assessment of patients, respiratory assessment and respiratory rate measurements became the main and most important measurements. This aspect, which emerged from the nurses’ interviews, was confirmed through the patient and nurse surveys and the electronic health records, which identified the same three main needs, including breathing. However, disagreement was found regarding the need for sleep and rest, which was considered important by nurses and patients but appeared less in the electronic health records.

Full agreement was found between the different data sources with regard to breathing and communication needs and related to new interventions (such as pronation, high-flow oxygen therapy with a Venturi mask, and video calls).

## 4. Discussion

In this study, the experiences and nursing care needs of patients with COVID-19, admitted to hospital with serious symptoms, were investigated and compared with the experiences and professional actions of the nurses who took care of them. To integrate these very different data, the joint display methodology was used to compare the agreements and disagreements obtained from the various data sources and to have a broad perspective of this complex phenomenon.

### 4.1. Nursing Care Needs (NCNs): Survey with Patients and Nurses

The need to breathe and need for communication/relationship were perceived as important by both patients and nurses, while there was a difference with the need to sleep and rest, as well as for maintaining cardiovascular function. For patients, the difficulty sleeping, in addition to not letting then recover energy, made the nights very long and facilitated the onset of feelings of fear and even anguish. The nurses noted this need, but from the interviews it was not clear whether there was a clear awareness of this possible discomfort in some patients during the night, which they often did not make explicit.

In the nurses’ responses, there was attention given to keeping various needs and many potential problems of the patients under control, as well as ensuring the necessary diagnostic and therapeutic procedures were conducted. Their attention was mainly aimed at ensuring vital functions (i.e., breathing and circulation), but in any case most of the needs indicated by Cantarelli’s model [30] received an importance rating equal to or greater than 8/10. Relatively lower scores were obtained for some basic needs (i.e., movement, nutrition/hydration, hygiene). The needs for which there was a greater relative difference in assessing their importance between nurses (higher) and patients (lower) included: uro-intestinal elimination, environmental safety, and maintenance of an adequate cardiac and circulatory function. Patients were aware of the importance of breathing when it failed, but they had difficulty evaluating the importance of their cardio-circulatory function, due to a lack of knowledge that nurses had. The vision of the nurses, being professional, was necessarily broader than that of the patients, since they had to deal with possible problems that the patients were still not yet able to perceive. They had a broader vision, but which may not have been sufficient to perceive the depth of the patients’ discomfort/suffering, especially in the absence of valid communication/relationship, as in the case of the need for sleep and rest.

### 4.2. Healthcare Interventions: Survey with Patients and Nurses

With regard to the healthcare interventions, by cross-referencing the frequency of the responses with the assessment of the level of importance, we found some similitudes and differences between the assessments made by the nurses and those made by the patients.

The interventions that were considered important by many patients were congruent with the needs they expressed (i.e., breathing and communication). They considered receiving oxygen and having clear information on their situation very useful. Regarding the need for sleep, approximately half of the patients rated positively the effort made by the nurses to create a peaceful and quiet nighttime environment. Even though not expressed as significant needs, being informed about how to prevent the spread of COVID-19 (i.e., ensuring a safe environment) and receiving blood samples and treatments to maintain cardiovascular function were greatly appreciated by the patients.

The interventions reported by the nurses were quite consistent with the detection/assessment of patients’ needs made by the nurses. In fact, there were many that included all the needs highlighted in Cantarelli’s model [30], and almost all were rated with a score of ≥8/10. Only one intervention, regarding the promotion of mobilization, received a score of 7. As regards breathing, in addition to the administration of oxygen (i.e., therapeutic procedures), nurses rated as “very important” the assessment and monitoring of respiratory signs and the management of anxiety/dyspnoea attacks. Regarding communication, nurses believed it was important to maintain a constant presence at the patient’s bedside and to encourage communication with their family members. In addition to guaranteeing the necessary therapeutic procedures, including oxygen therapy, the surveillance of vital signs, the level of consciousness, and the correct use of personal protective equipment were also considered very important by nurses.

Nurses were also asked which new interventions they had to implement to address the needs of patients with COVID-19. Among these, a small percentage of nurses reported pronation, the adoption of innovative communication strategies (i.e., video calls), and the use of oxygen supply devices (i.e., Venturi system, CPAP). These were not entirely new activities, except for those who had no previous experience in intensive care units, but they were performed with a frequency and in conditions that were very different compared to in the past.

### 4.3. The Nurses’ Experiences of the Situation: Data from Nurse Interviews

Facing the reality was psychologically traumatic. Nurses were by no means prepared to handle such a situation and of that extent (you won’t believe it if you don’t see it). Nurses perceived the fear in the patients’ eyes and felt a profound sense of helplessness in the face of the rapid evolution of the disease and the many deaths, with many deaths of people alone without the comfort of the presence of their loved ones, as already reported in the literature [5,41,42,43,44,45]. Despite the dramatic nature of the situation, for some nurses it was also a positive experience, as it enriched them at a human and professional level. This also emerged in the study by Falcò-Pegueroles et al. [46], where the pandemic was also described as an opportunity for professional development. Over time, however, tiredness set in, and the staff’s personal availability partially decreased. Especially in the initial stages of the pandemic, there was a lot of fear of being infected and of infecting family [44].

Over time, maintaining intense relationships with patients with very uncertain, if not terminal, clinical outcomes was difficult for nurses to cope with. Some stories of patients with whom there had been a prolonged or a closer human and professional relationship remained vivid in the nurses’ memory; with great joy when the outcome was positive and great suffering when the patient died. Many patients dying in solitude generated anguish and made nurses think more or less consciously about the idea of death and its widespread presence in that context. Their pain was so profound at times that nurses wondered if it was worth continuing to be a nurse, exhibiting a high risk of burnout. The idea of death hovered over all healthcare workers, to which they reacted with ambivalent behaviors. In fact, there was often the tendency to become insensitive to the fate of patients and focus on technical actions. On the other hand, accompanying patients until they died in solitude, with closeness and bodily contact, made it difficult for nurses not to empathize with them. Being close to the dying patient was also a profound need of the nurses themselves [5,44]. This closeness with patients also made it difficult to passively accept the rules in the department that obliged nurses not to resuscitate serious patients, against which some nurses tried to rebel, knowing that they no chance of changing things but felt they should advocate for those patients. This was just one of the many examples of ethical conflicts or moral distress experienced by nurses and other healthcare workers during the pandemic [5,44,45].

What seemed to have helped nurses manage their personal difficulties was the strong motivation to be involved, the awareness of their social role, and the desire to make their contribution against the terrible pandemic. Having solid professional experience also helped nurses to better manage their emotions [46].

### 4.4. The Professional Role: (Data from Nurse Interviews)

From the interviews, we found that a nurse’s professional role is multifaceted and requires extensive knowledge and good communication-relational skills. Nurses bring added value by being very close to patients, for many hours a day, making them feel their closeness and respect. These aspects were also highlighted by Virginia Henderson in one of her last studies [47]. The humanization of care and the profound respect for the dignity of the people nurses care for emerged as a priority for many colleagues. This is in line with Watson’s theory of “human caring” [48]. Nurses in this context still paid a high price in terms of emotional and moral stress, with situations of ethical conflict as also reported in other studies [5,44]. It should be noted that even in the case of the nursing care provided to patients with COVID-19, basic nursing care and fundamental care (e.g., hygiene, elimination, mobility, etc.) was practiced but usually poorly described or documented, almost as if such treatments are taken for granted and perhaps even given little consideration. The need to return to the origins and give meaning and dignity to these treatments was expressed by nurses in a few interviews, but this is a current issue and is linked to the thoughts and actions of other colleagues internationally [49].

### 4.5. Facilitators and Barriers: Data from Nurse Interviews

The context in which nursing care was provided was very difficult from a clinical and organizational point of view. To describe the situation, not surprisingly, some nurses used war metaphors (e.g., going to the front), as also highlighted in other studies [45]. The normal procedures and ways of providing nursing care, and in general all healthcare procedures, were suddenly disrupted due to the need to rapidly find new solutions, with approaches initially heavily based trial and error. Improvised multi-professional teams, with different levels of competence and experience, who had never collaborated with each other before, suddenly found themselves facing an unknown disease often with sudden serious or fatal outcomes [45].

Many factors facilitated nurses, especially those on the front line, in managing the difficult situation: everyone being in the same situation; perceiving the support of the population (initially); and the climate of discussion, collaboration, and mutual help that was immediately established in most teams. Other facilitators included the support of local management (i.e., heads of departments), having nurses experienced in intensive care in each team, and the reduction in the distance from other professionals, especially with physicians [45]. The importance of the strategic role of healthcare management has also been reported in other studies [50,51].

Factors experienced as barriers to the quality of care instead included the necessary standardization of many activities (i.e., giving at least the minimum to everyone) and the lack of adequate intensive care skills of many nurses, who could only support the work of their more expert colleagues and were not directly in charge of critically ill patients. This situation was a source of psychological stress and moral distress [5,44]

In this experience, we also observed the link that exists between hard factors (i.e., the qualitative and quantitative composition of the teams) and soft factors related to healthcare work (i.e., individual motivation, social climate, and leadership styles) and their contribution to the quality of the final results, hence the importance of the work carried out by professional and company management, who had to invent new and rapid technical-organizational solutions, sometimes with a trial-and-error approach and sometimes by just being creative. The importance of these factors has also been highlighted by several other studies [52,53].

### 4.6. Patients’ Evaluation of Nursing Care: Data from Questionnaires and Patient Interviews

Most patients (91.5%), in general, rated the nursing care they had received as “good/excellent” (with a score of 6 out of 7) and the interviews confirmed the results of the questionnaires. The nurses were well regarded, both for their competence and their communication-relational skills. Competence was assessed by observing certain elements such as knowing how to answer their questions, and efficiency and safety in the activities they performed. With regard to communicative-relational skills, the nurses’ ability to approach the patients’ bodies with sensitivity and attention, the humanity they showed, and their perception of an attentive attitude towards their clinical and personal conditions were appreciated by the patients. Some patients described them as their “guardian angels” and underlined the importance of continuity in the relationship and the willingness to listen. Some mild issues were expressed regarding limited freedom within the ward, the long length of time that sometimes passed between the patient’s call and the nurse’s arrival, and regarding the way in which nurses reassured patients’ relatives and friends.

### 4.7. Patients’ Experiences with Respect to Their Needs: Data from Patient Interviews

Patients had various experience regarding their needs/problems that were related to the course of the illness. They could experience discomfort/frustration with their body image, seeing themselves as unkempt, being dependent on the presence of others, and having difficulties communicating with nurses (also because of the PPE). They were very concerned about their family members and wished to inform them about their condition. The fear of death was widespread and increased by what they saw happening to other patients in their room. As their symptoms became worse, patients felt the need to say goodbye to their loved ones, perhaps for the last time. Priorities changed, and patients focus on themselves and their destiny. The moment of intubation was perceived by some as a probable journey of no return. During the pre-intubation phases, patients had little time to accept it, and this is often when they became aware of the seriousness of their condition. Another critical moment was the awakening phase, for which some reported being unprepared due to a lack of preliminary information, finding themselves immobilized and unable to speak due to being intubated, and therefore experiencing sensations of fear and disorientation. The external context was also a source of fear: the presence of the military and the civil protection tents contributed to creating a surreal, warlike scenario. All these aspects were also described in other studies [41,42,43].

Having to depend on nurses for daily life activities, and especially for personal hygiene, generated feelings of frustration and helplessness in patients, even if these activities were acknowledged to have been managed with professionalism and sensitivity by nurses. With the improvement in their clinical conditions, a self-motivating factor for patients was the recovery of autonomy in carrying out some daily activities and the hope of still having “a future” (i.e., going back home and resuming hobbies and relationships) [41]. It was also very useful to maintain contact with family members via telephone and video-calls, as well as having another patient in the room to spend some time and chat with. The latter aspect, however, was experienced negatively during the acute phase of the disease due to the perception that the other patient was going to die.

### 4.8. Nursing Care (Data from Questionnaires and Interviews with Nurses and from Clinical Records)

Through the various sources (i.e., questionnaires, interviews, and clinical records) we attempted to reconstruct the professional approach used by nurses in the care they provided to patients with COVID-19. We explored if any reference had been made to a precise and conscious professional model and to a specific problem-solving methodology or any other model. From the data we collected, it is not clear if a specific conceptual model of nursing care was used, not even that of Marisa Cantarelli’s Model of Nursing Performance [30], which was used as a guide for the development of the questionnaire and therefore could have led some of the respondents to take it into consideration when describing their professional practice. Some of the model factors could be distinguished, although they might have been used unknowingly [47,54].

It should be considered that the nurses involved in the study came from different training backgrounds and sometimes even from different countries. The prevailing approach was that of professionals who, standing alongside patients, responded to their needs/problems in a responsive way, using both intuitive thinking and experience, an aspect also highlighted in other studies [55]. With this method, the nurses, especially if they were experienced, managed to integrate all of their knowledge (i.e., theoretical, experiences, and information received from other professionals) and adapt it to the specific clinical-care evolution of their patients and, when possible, also according to their patients’ will. This flexibility and ability to adapt, in addition to a competent and respectful closeness, were some of the strengths of the nursing care provided in this difficult and complex situation.

The priority assessment activities were therefore aimed at controlling and supporting breathing and vital functions. Respiratory weaning was one of the most relevant healthcare problems and required great efforts and solid skills, typical of intensive care nurses. Of all the needs, the one that was least reported was sleep and rest, which was considered important by patients but probably less so by nurses. Nurses should place more attention on this need and act accordingly [8]. Among the healthcare interventions, those rated as “very complex” concerned knowing how to detect a patient’s clinical deterioration early and knowing how to communicate with patients and family members in particularly difficult situations and during a crisis.

Professional objectives were often present and guided professional action but were mainly indicated in very general terms. On other occasions, they were not made explicit or written but were deduced from the interventions that nurses implemented. Personal objectives were indicated in a very specific way and related to precise professional actions and evaluations only in few cases.

Among the many healthcare interventions, during their interview, nurses placed great importance on their relationship/communication with their patients, especially in certain moments that patients perceived as critical (i.e., the pre-intubation and the awakening/respiratory weaning phase). Communication was particularly problematic with patients who were conscious but intubated or tracheotomized, while, in the less critical phases, offering patients the possibility to communicate with their loved ones via telephone and video-calls was considered very useful. This urge to communicate, previously highlighted in other studies [56,57], became even stronger during the pandemic, due to the absence of relatives and the difficult working conditions. The relationship with patients was emotionally burdensome, especially if they were seriously ill or dying. With the latter, even if they were unconscious, nurses tried to communicate with words and gestures, almost acting as a bridge with their families and partially carrying out some vicarious functions (i.e., proximity, physical contact, accompaniment). The nursing maneuvers required for COVID-19 patients were not completely new to intensive care nurses. What was new was having to do them in such difficult conditions, so frequently (e.g., pronation), and with the support of colleagues who did not have intensive care skills.

The outcomes of nursing care were not described in a precise and timely manner, in line with what was reported for the objectives. Therefore, the outcomes were often described in very general terms. The issue of not defining professional outcomes is still widely debated among professionals and in the literature [58].

### 4.9. Limitations

This study was conducted in only one hospital and therefore may have been affected by socio-organizational and cultural-professional characteristics relevant only to the local context.

For the quantitative part, the response rate to the survey was not high, especially among nurses (less than 20%), while for the patients, the response rate was higher than 30%. The questionnaire developed considering Cantarelli’s nursing performance model of care [30] was only used descriptively, so we did not test reliability. Although we considered the entire eligible populations, we did not calculate a sample size for the nurse and patient samples. This can be considered a limitation for the generalizability of the results. 

The choice to conduct a mixed-method study, on the one hand, enabled us to obtain an overall perspective of the phenomenon under examination and, on the other hand, made it more complex to analyze and synthesize data drawn from different sources. In order to be able to draw up a summary and achieve a clear overall view only some of the many data and themes present have been highlighted in this paper, potentially underestimating others that may have been significant or relevant.

## 5. Conclusions

This study offered the opportunity to highlight how nursing care was provided during the COVID-19 pandemic in our context. Patient and nurse perceptions of the needs were very similar for certain aspects. In this difficult context, nursing care was complex, multifaceted, and with a constant and competent technical-relational presence required at the patients’ bedside. Nurses showed how they could be very creative and resilient, and they were able to find coping strategies to deal with this extraordinary situation, although in the long term, high levels of tiredness were experienced. 

The prevailing approach was that of professionals who, standing alongside patients, responded to their needs/problems in a responsive way, using both reflective-intuitive thinking and experience. With this method, nurses, especially if experienced, managed to integrate knowledge from different sources (theoretical, ethical, experiential, and other professionals) and adapt it to the clinical-care evolution of their patients and, when possible, also to their wishes. This flexibility and ability to adapt, in addition to a competent and respectful closeness, were some of the strengths of nursing care in this difficult and complex situation. However, the definition and evaluation of the outcomes of nursing care need to be improved.

As already reported in the literature, nurses really struggled when seeing their patients die in solitude. Patients were also frightened and distressed due to the impossibility of seeing their relatives and owing to their own condition and the condition of other patients. As in other studies, nurses suffered psychological distress and ethical conflict. Patients evaluated the care they received as very satisfactory. 

This study provides useful information for future situations like the COVID-19 pandemic and can be useful for management and health policies. Nurses were the backbone of the health systems globally during the pandemic and therefore should be recognized and valued for their strategic role.

## Figures and Tables

**Figure 1 healthcare-12-00114-f001:**
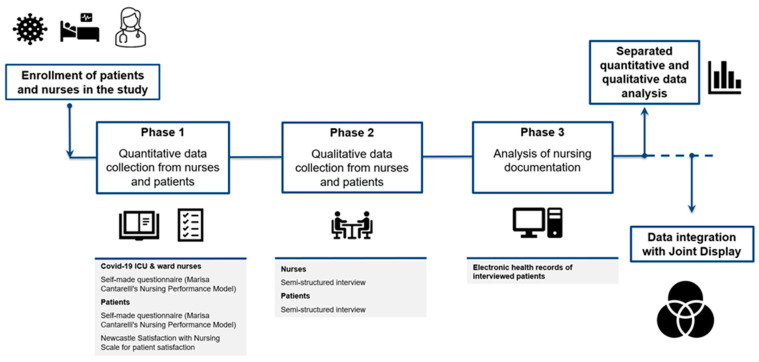
Study phases overview.

**Figure 2 healthcare-12-00114-f002:**
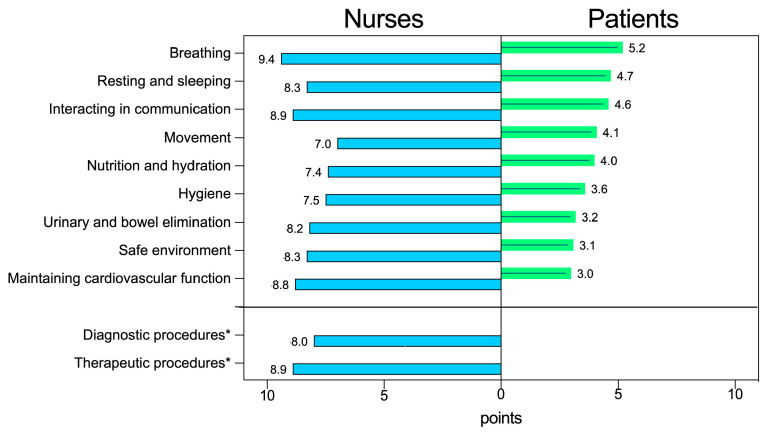
Importance attributed by nurses and patients to each need according to the Cantarelli’s nursing performance model (NPM) [30]. Values indicate means. * Needs explored with nurses only.

**Figure 3 healthcare-12-00114-f003:**
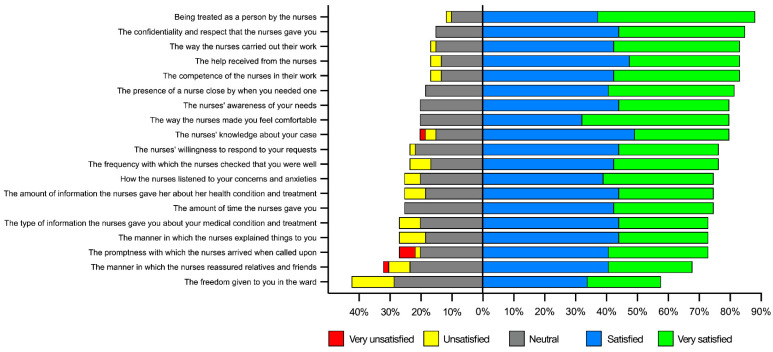
Level of patient satisfaction with nursing care during the COVID-19 pandemic.

**Table 1 healthcare-12-00114-t001:** Fundamental nursing needs based on Marisa Cantarelli’s Nursing Performance Model [30].

Marisa Cantarelli’s Nursing Performance Model [30]
Fundamental nursing needs
Breathing
2. Nutrition and hydration
3. Urinary and bowel elimination
4. Hygiene
5. Movement
6. Resting and sleeping
7. Maintaining cardiovascular function
8. Safe environment
9. Interacting and communication
10. Need for the therapeutic procedures *
11. Need for diagnostic procedures *

* Not included in the patients’ evaluation.

**Table 2 healthcare-12-00114-t002:** Baseline characteristics of nurses and patients.

Nurses’ Characteristics	All Nurses *		Patients’ Characteristics	All Patients	
	(*n* = 100)			(*n* = 57)	
Age			Age, mean years, ±SD	69	±11.0
20 to 30 years	33	(34.4)	Female, gender	18	(31.5)
31 to 50 years	46	(47.9)	BMI, mean kg/m^2^, ±SD	27.4	±5.5
51 to 60 years	17	(17.7)	Autonomous/partially autonomous	55	(96.4)
Female, gender	68	(70.8)	Previous comorbidity	32	(56.1)
Years of experience as nurse			Hypertension	17	(29.8)
0 to 5	24	(25.0)	Diabetes mellitus	9	(15.7)
6 to 20	47	(49.0)	Renal impairment	3	(5.2)
over 20	25	(26.0)	Heart diseases	9	(15.7)
Post-graduate education	80	(80.0)	Other	7	(12.2)
Intensive care employment during pandemic	33	(34.7)	High educational level	19	(33.3)
			Hospital length of stay during COVID-19 pandemic, median (IQR)	17	(13–37)
			Admission in ICU	25	(43.8)

Values *n* (%), unless otherwise specified. ICU = intensive care unit, BMI = body mass index; COVID-19 = coronavirus disease 2019; SD = standard deviation; IQR = interquartile range. (*) Four cases are missing.

**Table 3 healthcare-12-00114-t003:** The most common nursing intervention for the nine Cantarelli’s nursing performance model (NPM) [30] nursing needs according to nurses’ assessment and patients’ perceptions.

NURSES (*n* = 100) Top Three Most Common Nursing Interventions Performed					PATIENTS (*n* = 57) Top Three Common Nursing Interventions Received				
Interventions	*n*. (%)		Degree of Usefulness		Interventions	*n* (%)		Degree of Usefulness	
Breathing									
Assessment/monitoring of respiratory parameters	99	(99.0)	10	(9–10)	Supply me with oxygen	49	(85.9)	8	(5–10)
Prevention of complications from devices used	95	(95.0)	8	(7–10)	Airing out the room	37	(64.9)	6	(4–7)
Support in cases of anxiety/dyspnoea	91	(91.0)	9	(8–10)	Getting me into a sitting position	27	(47.4)	6.5	(3–8)
Resting and sleeping									
Discretionary use of drugs in reserve	85	(85.0)	8	(6–10)	Promoting a peaceful and quiet environment	32	(56.1)	8	(5–9)
Communication and relationship for anxiety situations	83	(83.0)	8	(7–10)	Administering therapy	28	(49.1)	7	(4–10)
Promoting a comfortable sleeping environment	83	(83.0)	8	(5–10)	Switching off or dimming the light	27	(47.4)	7	(4–10)
Interacting in communication									
Maintaining a constant presence at the patient’s bedside	94	(94.0)	8	(6–10)	Provide me with clear information about my situation	45	(78.9)	8	(6.5–10)
Maintain appropriate communication	92	(92.0)	9	(8–10)	Answer my questions	39	(68.4)	8	(6–10)
Ensuring communication with relatives	87	(87.0)	10	(8–10)	Explain to me which interventions they were going to do	38	(66.7)	8	(7.5–10)
Movement									
Help with moving without aids	85	(85.0)	7	(5–9)	Getting me to exercise	28	(49.1)	8	(5–10)
Promotion of physiotherapy programme	83	(83.0)	8	(5–9)	Helping me get out of bed, sit in an armchair, and get back into bed	21	(36.8)	8	(4–10)
Maintaining motor skills	80	(80.0)	8	(6–10)	Helping me move in bed	20	(35.1)	7	(4.5–8.5)
Nutrition and hydration									
Monitoring of glycaemia (for diabetic patients)	99	(99.0)	9	(8–10)	Checking my glycaemia	34	(59.6)	7	(3–10)
Surveillance of nutritional status and hydration	94	(94.0)	8	(7–10)	Asking me what I preferred to eat	33	(57.9)	8	(6–10)
Surveillance and management of disorders such as nausea and vomiting	89	(89.0)	8	(7–10)	Giving me drips	32	(56.1)	8	(5–10)
Hygiene									
Skin surveillance and/or care	95	(95.0)	9	(8–10)	Provide me with hygiene care supplies	37	(64.9)	7	(3–9)
Oral cavity surveillance and/or care	94	(96.0)	9	(8–10)	Help me wash or shower	26	(45.6)	6.5	(4–10)
Partial or full help with hygiene	91	(91.0)	8	(7–10)	Accompany me to the bathroom	24	(42.1)	7	(4–10)
Urinary and bowel elimination									
Control on adequate elimination	95	(95.0)	9	(8–10)	Accompany me to the bathroom	20	(35.1)	7	(4–10)
Bladder catheter management	93	(93.0)	8	(7–10)	Bring me the bedpan or the urinal	18	(31.6)	7.5	(2–10)
Assisting in the use of elimination aids (urinal, bedpan, commode chair)	80	(80.0)	8	(7–10)	Placing me in the commode chair	13	(22.8)	7	(0–10)
Safe environment									
Correct use of protective equipment	90	(90.0)	10	(7–10)	Explain how to prevent the spread of COVID-19	38	(66.7)	8	(7–10)
Preventing situations of disorientation and delirium	88	(88.8)	8	(7–10)	Maintain a clean environment	35	(61.4)	8	(7–10)
Preventing accidental falls by acting environment	84	(84.4)	8	(7–10)	Explain how to avoid falls and injuries	19	(33.4)	8	(5–10)
Maintaining cardiovascular function									
Surveillance of vital parameters	95	(95.0)	10	(9–10)	Take my blood pressure	46	(80.7)	7	(6–10)
Surveillance of states of consciousness	95	(95.0)	10	(8–10)	Take my body temperature	45	(78.9)	7	(6–10)
Maintaining an adequate body temperature	95	(95.0)	8	(8–10)	Administer my therapy	39	(68.4)	8	(6–10)
Therapeutic procedures *									
Therapy administration and monitoring	96	(96.0)	9	(8–10)	n.a.				
Administering oxygen therapy	94	(94.0)	10	(8–10)	n.a.				
Monitoring possible complications due to therapy	94	(94.0)	10	(8–10)	n.a.				
Diagnostic procedures *									
Collecting biological samples	95	(95.0)	8	(5–9)	n.a.				
Monitoring disease evolution	93	(93.0)	9	(8–10)	n.a.				
Supporting patients with anxieties and fears	85	(85.0)	8	(5–10)	n.a.				

* Unexplored items for patients. The values are shown as frequencies and percentages, while the degree of usefulness are medians with a 25th and 75th percentile.

**Table 4 healthcare-12-00114-t004:** New interventions adopted by nurses during the COVID-19 pandemic for each of the nine needs of the nursing performance model (NPM).

NPM Nursing Needs	Novelty Nursing Interventions	*n*. (%)	
Breathing	Use of oxygen delivery devices (Venturi system, CPAP)	7	(7.0)
Resting and sleeping	Use of new medicines for analgesia-sedation due to depletion of usual medicine stocks	4	(4.0)
Interacting in communication	Ensuring communication with relatives (telephone, tablet, or other)	10	(10.0)
Movement	Implementing pronation	13	(13.0)
Nutrition and hydration	Feeding tracheostomised patients	2	(2.0)
Hygiene	Teaching about hygiene problems for home discharge	2	(2.0)
Urinary and bowel elimination	Handling fecal incontinence devices	2	(2.0)
Safe environment	Use of personal protective equipment	5	(5.0)
Maintaining cardiovascular function	Invasive monitoring of haemodynamic parameters	3	(3.0)
Therapeutic procedures	Artificial ventilation in patients with respiratory insufficiency	2	(2.0)
Diagnostic procedures	Monitoring of disease evolution COVID-19	4	(4.0)

NPM = Nursing Performance Model; CPAP = Continuous Positive Airway Pressure; COVID-19 = Coronavirus Disease 2019. Values indicate frequencies (percentages).

**Table 5 healthcare-12-00114-t005:** Macro themes and themes from nurses’ interviews.

Nurses Interviews (*n* = 16)
Macro Theme	Themes
The context	1.1The clinical/professional context1.2The organizational context
2.Nurses’ experiences and emotions	2.1You won’t believe it until you see it2.2It was a nice experience2.3Your own emotions2.4Confrontation with death2.5Your own resources2.6I no longer want to be a nurse




3.Facilitators and barriers to patient care	3.1Interprofessional collaboration3.2Support from superiors3.3Being experts3.4Lack of specific competencies3.5Standardization of care



4.Nursing care	4.1The nursing care approach to COVID-19 patients4.2Assessment of patients and their needs4.3Professional objectives and priorities4.4Professional performance4.5The outcomes of nursing care



5.The professional role	5.11A multifaceted role5.2Fundamental care


**Table 6 healthcare-12-00114-t006:** Macro themes and themes from patient interviews.

Patient Interviews (*n* = 15)	
Macro Theme	Themes
My needs/problems	1.1 I had no information or it was incomplete 1.2 Putting all the pieces of the jigsaw puzzle 1.3 Physical problems 1.4 Psychological problems 1.5 Communication/relational problems

2. My emotions	2.1 Worry and fear 2.2 Frustration and helplessness 2.3 Priorities that changed

3. What helped me	3.1 Internal factors 3.2 External factors

4. Nursing care	4.1 Caring activities 4.2 Professionalism 4.3 How nurses are perceived



**Table 7 healthcare-12-00114-t007:** Joint display integrating quantitative and qualitative findings from questionnaires, interviews, and clinical records [1].

Quantitative Results (Nurse Survey)	Clinical Records	Quantitative Results (Patient Survey)	Qualitative Results (Nurse Interviews)	Qualitative Results (Patient Interviews)	Agree	Disagree	Integration of Quantitative and Qualitative Data.
		78% of the nurses were aware of their needs. 80% of the patients were fully satisfied with the nursing care received.	Many patients were grateful for the care received, also because they realized the nurses’ high level of stress and dedication to ensure the best possible care.	Appreciation of the nurses’ technical and professional competence.	X		The nursing staff is fully committed to ensuring the highest quality of care and patients realize this.
Needs considered important by nurses: Breathing Sleep and rest Interaction in communication	Need to breathe reported in most clinical records. The need for sleep and rest was reported less frequently. Management of crisis conversations with patients and family members	Needs considered important by patients: Breathing Sleep and rest Interaction in communication	Importance of the need to breathe, respiratory monitoring, and timely interventions. Importance of the need for sleep and rest (in relation to asthenia, disturbed sleep). Communication managed by nurses at different moments (admission, to reassure, support during respiratory weaning, reassurance during functional recovery).	Breathing difficulty made the person experience the fear of dying. Importance of the need for sleep and rest. Support and encouragement from nurses. Difficulty recognizing faces due to masks, making communication difficult.	X X X		The same perception of the main problems was identified both from the patients’ and nurses’ points of view. The need for breathing is not mentioned in the patient interviews. The need for sleep and rest was described as as being important, but appeared less in the clinical records.
	Implicit or rarely stated objectives		Generic objectives: generic objectives were pursued based on clinical evolution and changes in knowledge about the disease.		X		
34% of nurses reported that they had implemented new actions to address the need to breathe (pronation, high flow oxygen therapy with Venturi mask) and for communication (video calls).	Interventions related to the need to breathe and communicate were present in most of the clinical records.	Giving me oxygen (Median 8 (5–10]) Letting me communicate with my family via mobile phone/tablet (Median 8 (6.5–9.5)). Explaining to me the interventions they were about to perform (Median 8 (5.75–10)).	New skills emerged in relation to breathing and communication needs. New actions: high flow oxygen therapy, pronation, use of new scores—early warning score, use of technology to facilitate communication.	Facilitating contacts with family members (mobile phones/video calls). Clear information about nursing interventions.	X		There is full agreement between different data sources about breathing and communication needs and the related interventions.
		Giving me clear information about my situation (Median 8 (6.5–10)).		Insufficient information about the clinical situation, intubation, and awakening.		X	There are differences between the quantitative and qualitative data of the patients.
			Seeing them all naked in these big rooms. Distorted faces.	Body image—seeing themselves unkempt caused embassment and suffering.	X		
			For nurses, there was not enough time to provide care due to the workload.	Time that passes slowly causes suffering		X	
			Continuous patient turnover. Nurses suffered seeing people die alone.	Hearing what happened to other patients increased the fear of death.	X		Anguish for death, even if experienced differently by nurses and patients.
			Sense of helplessness and fear of not doing the right thing.	Feeling helpless for not being able to do anything anymore.	X		Sense of helplessness for both stakeholders, but for different reasons.
			Fear of contagion.				
			Nurses had difficulty capturing sudden changes due to poor perception of patients’ respiratory deterioration.	They felt confused and surprised when they were unexpectedly and suddenly told that they would be intubated.	X		

## Data Availability

The data that support the findings of this study are available on request from the corresponding author (L.B.).The data are not publicly available due to privacy restriction.

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
