# Peer review of "Nurses Response to the Physical and Psycho-Social Care Needs of Patients with COVID-19: A Mixed-Methods Study"

_healthcare, 2024, doi:10.3390/healthcare12010114_

Round 1

Reviewer 1 Report

Comments and Suggestions for Authors

Comments on the Quality of English Language

Overall, the manuscript is well written. Needs a minor revision of the English.

Author Response

Dear Reviewer,

Thank you very much for your comments, which helped us to improve the manuscript. In the file enclosed the answers to your suggestions. All changes in the paper are in red. 

Reviewer 2 Report

Comments and Suggestions for Authors

Dear Authors,
thank you very much for the opportunity to read your work. Research proposals with a mixed design are of great interest because of the approach from different research paradigms. The main difficulty consists in the integration of both methodologies and knowing how to capture this aspect in the manuscript, but I consider that you have managed to capture it in the manuscript. However, in Figure 1 Study phases overview, it is necessary to represent the integration process that has been carried out (Data triangulation (expressed as data triangulation is insufficient, it would be better to also point out the concept of integration).

The study should conform to the reporting criteria of the STROBE guideline for observational design and COREQ guideline for qualitative design.

In the abstract, the content of the methodology followed in each of the phases should be expanded in greater detail to reflect the scope of the design.

Keywords: "Experience" is a very unspecific descriptor term (it was difficult for me to check if it really is a MeSH term because entering it in the MeSH Database retrieves 86 possible terms). "Mixed method" is not a MeSH term, another term should be used to refer to the research design. "Patients' needs" is not a MeSH term. The correct MeSH for "Intensive Care Unit" and "Pandemic" are in plural ("Intensive Care Units" and "Pandemics".

Introduction: In the first sentence it is convenient to indicate the year of the WHO report, since the last year will become out of date in the years following the publication of the manuscript.

Page 2 (line 66) aclarate the acronims for: SARS, MERS, H1N1, EBOLA.

Methods:

The design, participants and sample, inclusion and exclusion criteria, data collection, data analysis and ethical criteria should be better structured in each  3 phases. In the quantitative phase also: instruments, sociodemographic variables and clinical variables. In the qualitative phase also: methodological approach, characteristics of the investigators, methodological rigor. It should be specified for each study population (nurses, patients and patient clinical records).

Participants: The participants section should be better explained, since the sampling and selection of participants is not the same for the qualitative design as for the quantitative design. In my opinion, it is better to include the participants section in each of the phases of the research to point out the characteristics of the sample in each of the designs. No selection criteria (inclusion, exclusion criteria) are specified in either of the three phases of the research.

Quantitative study phase:

Page 3 indicates in wrong order Table 2 (line 117) and Table 1 (line 125), thus the Table 2 is missing into the text. In my opinion, the table is irrelevant and unnecessary, and the information it contains can be described in the text.

Data collection: The questionnaires used were developed ad hoc? point out this aspect.

Sample: The sampling carried out is not described with sufficient precision for either of the two phases. For the quantitative phase, an estimate of the required sample size should be provided. The authors report using a non-probabilistic sample for the quantitative phase, mentioning within this phase that in the questionnaire the individuals in the sample were asked to indicate their agreement to participate in the qualitative phase, but a methodological problems apply: no describe the criteria for selection ("sampling") process for the qualitative study within the quantitative phase. It is not explained how it was ensured that the individuals who responded to the questionnaire were nurses belonging to the target population (the target population and the eligible population are not described either).

Qualitative study phase:

COREQ items included in domain 1 (research team and reflexivity) and 2 (study design) should be described in more details. Available in: https://www.equator-network.org/reporting-guidelines/coreq/

The qualitative design does not describe the methodological and theoretical approach followed.

There is a third phase of the study that also has a quantitative design (2.5 Analysis of the Nursing Documentation). This phase of the study is understood to be a quantitative design for the evaluation of patient clinical records, which should be explained in the methodology including the respective sections (in Figure 1 it is Phase 3?).

Results:

The presentation of results should follow the same structure (same phase headings) as in the methodology so that readers do not lose the common thread.

The sentence: "Of the 560 nurses we invited"... It was not explained in the methodology that a sample size was estimated in order to justify inviting 560 nurses (was it necessary to invite 560 nurses in order to obtain statistical significance?).

the authors mix the presentation of results with nurses and patients, it would be easier to understand if they first describe the results of a population and in another section those of the patients. Do not mix results from the two populations.

If possible, insert tables and figures in the text immediately after they are cited (e.g., figure 2 is cited before table 3, but table 3 is presented before figure 2).

In Table 2, the first column of Table 2 shows results for nurses, while in Table 3 the first column corresponds to patients.

The same with table 4, which is very lost in the middle of the other tables and figures.

Patient interviews (Qualitative phase?):

The results are also not structured to clearly distinguish the results in the two populations (nurses and patients).

A table to describe the characteristics of the informants (nurses and patients) is missing. They also need to include a table showing the categories of analysis and themes identified.

Results from the clinical patients records (Phase 3?). I understand that it is complex to extract and organize information from patients records, but this process should be standardized as closely as possible. Table 5 should be more user friendly.

Integration of the qualitative and quantitative findings

Why do you divide table 6? Is there any criterion for them to be two tables? In table 6, the conclusion column is not correct. It is not a result, but an interpretation of the researchers. To reach an agreement on the results, it would be necessary to use some technique of expert consensus (this would imply another phase in the mixed design).

Discussion:
To follow the logical order, the discussion should be conducted following the above structure of phases.

Author Response

(The authors gave the same response as above.)

Round 2

Reviewer 2 Report

Comments and Suggestions for Authors

Dear authors, in this second review you have improve some recomendations, but some aspects need to be improved.

Abstract: in the results subheading you describe five themes in nurses' interviews: The quotation marks for the first topic are missing, and the "and" before the last topic is missing. 

Keywords: "Quantitative Research" is not a MeSH term.

Introduction: Correct.

Methods:

Participants: Although you have included the entire population, it is methodologically correct to estimate a minimum sample size necessary to know the estimated necessary sample proportion (but this is a methodological issue that does not affect the validity of the sample obtained).

Data collection: include in the text that is an "ad hoc" questionary.

Data analysis: Where say "...continuous variables by means (SD)..." should describe "...by means and standard deviations (SD)..."

Review the numeration of the sub-headings; for example in 2.3. Qualitative study phase:  you describe 2.3.1 Qualitative approach; and repeat 2.3.1 Researcher characteristics and reflexivity.

Results: To follow the same order in the exposition of results as outlined in the methodology, you should describe in the title of Table 2 and in the results of Table 2 first the results of the nurses and secondly those of the patients. To follow the orderly presentation of results, in all the results (figures and tables) you should present the nurses' results first and the patients' results second.

Table 5 should be deleted. It is unnecessary, since it does not provide substantial information that needs to be shown in a table; it is sufficient to write this information in the text of the manuscript. The same for table 8.

On page 24 (line 867-869) review the line break in the heading.

The table on page 24 is wrong in numbering. It does not correspond to number 6. In my opinión, this table would be better displayed if the page is changed in horizontal.

Discussion: In the discussion it is more correct to follow the same criteria for numbering headings and subheadings when applicable.

The discussion should also follow the same order: nurses first, then patients.

Author Response

Dear Reviewer,

Thank you for your suggestions, which helped us to improve the paper. Now it is more logical, and the typos have been corrected. Please, find in the file enclosed the answers to your comments. All the changes in the paper are in red.
